# Spray-Drying Microencapsulation of High Concentration of Bioactive Compounds Fragments from *Euphorbia hirta* L. Extract and Their Effect on Diabetes Mellitus

**DOI:** 10.3390/foods9070881

**Published:** 2020-07-04

**Authors:** Ngan Tran, Minh Tran, Han Truong, Ly Le

**Affiliations:** School of Biotechnology, International University—Vietnam National University, Ho Chi Minh 700000, Vietnam; kimnganchemistry@gmail.com (N.T.); tramthiminh@gmail.com (M.T.); hantruong029@gmail.com (H.T.)

**Keywords:** *Euphorbia hirta* L., bioactive compounds, in vitro α-amylase inhibition, streptozotocin-induced diabetic mice, diabetes mellitus

## Abstract

The present study was performed to spray-dry the high concentration of bioactive compounds from *Euphorbia hirta* L. extracts that have antidiabetic activity. The total phenolic content (TPC) and total flavonoid content (TFC) of four different extracts (crude extract, petroleum ether extract, chloroform extract and ethyl acetate extract) from the dried powder of *Euphorbia hirta* L. were determined using a spectrophotometer. After that, the fragment containing a high number of bioactive compounds underwent spray-dried microencapsulation to produce powder which had antidiabetic potential. The total phenolic content values of the crude extract, petroleum ether extract, chloroform extract and ethyl acetate extract were 194.55 ± 0.82, 51.85 ± 3.12, 81.56 ± 1.72 and 214.21 ± 2.53 mg/g extract, expressed as gallic acid equivalents. Crude extract, petroleum ether extract, chloroform extract and ethyl acetate extracts showed total flavonoids 40.56 ± 7.27, 29.49 ± 1.66, 64.99 ± 2.60 and 91.69 ± 1.67 mg/g extract, as rutin equivalents. Ethyl acetate extract was mixed with 20% maltodextrin in a ratio of 1:10 to spray-dry microencapsulation. The results revealed that the moisture content, bulk density, color characteristic, solubility and hygroscopicity of the samples were 4.9567 ± 0.00577%, 0.3715 ± 0.01286 g/mL, 3.7367 ± 0.1424 Hue, 95.83 ± 1.44% and 9.9890 ± 1.4538 g H_2_O/100 g, respectively. The spray powder was inhibited 51.19% α-amylase at 10 mg/mL and reduced 51% in fast blood glucose (FBG) after 4 h treatment. Furthermore, the administration of spray powder for 15 days significantly lowered the fast blood glucose level in streptozotocin-diabetic mice by 23.32%, whereas, acarbose—a standard antidiabetic drug—and distilled water reduced the fast blood glucose level by 30.87% and 16.89%. Our results show that obtained *Euphorbia hirta* L. powder has potential antidiabetic activity.

## 1. Introduction

Diabetes mellitus is a chronic disease and the result of metabolic disorders in pancreas *β*-cells that have hyperglycemia [1,2]. Hyperglycemia is caused by a deficiency of insulin production by pancreatic (Type 1 diabetes mellitus) or the insufficiency of insulin production in the face of insulin resistance (Type 2 diabetes mellitus) [2,3]. Hyperglycemia causes damage to eyes, kidneys, nerves, heart and blood vessels [4]. According to the ninth edition, in 2019, of the IDF Diabetes Atlas released by the International Diabetes Federation (IDF), as of 2019, the total adult population living with diabetes in the age group of 20–79 years stands at 463 million, which is set to increase to 578 million by 2030 [5]. The present treatment of diabetes mellitus is focused on controlling and lowering the blood glucose levels in the vessel to a normal level [6]. Currently, there are six main classes of modern medicines used across the world for controlling blood glucose levels and two classes of injections [7]. The tablets are known as biguanides (metformin), sulfonylureas, thiazolidinediones (glitazones), meglitinides (glinides), *alpha*-glucosidase inhibitors and DPP-4 inhibitors. The classes of medications given by injection are incretin mimetics and insulin. However, most modern drugs have many side effects causing some serious medical problems during the period of treating. For instance, the main side effects of metformin are gastrointestinal on the initial state, including dyspepsia, nausea and diarrhea. The most common adverse effects with thiazolidinediones are weight gain and fluid retention, leading to peripheral edema and a twofold increased risk for congestive heart failure. Besides modern therapies, traditional medicines have been used for a long time and play an important role as alternative medicines [8]. According to the WHO, a plant-based traditional system of medicine is still the chief support of about 75–80% of the world’s population, mainly in developing countries having a diversity of plants [9]. Several patients in type 2 diabetes mellitus have used functional foods to reduce their blood glucose levels, such as olive leaf extract, turmeric and fenugreek.

*Euphorbia hirta* L. (*E. hirta*) is a common herb that belongs to the *Euphorbia* genus of the Euphorbiaceae family. *Euphorbia hirta* L. is found in pan-tropic, partly sub-tropic areas and worldwide including Australia, Western Australia, Northern Australia, Queensland, New South Wales, Central America, Africa, Indonesia, Malaysia, Philippines, China and India [10]. In Vietnam, *Euphorbia hirta* L. is commonly distributed in many provinces of the southern area. *Euphorbia hirta* L. was used as a traditional medicine in the treatment of diabetes a long time ago [10,11]. Furthermore, this plant contains a large number of phytochemicals including flavonoids, terpenoids, phenols, essential oil and other compounds containing antidiabetic potential [12]. Therefore, producing an ingredient containing *E. hirta* extract is necessary. However, phytochemicals such as flavonoids, terpenoids, and phenols are very sensitive to environmental conditions such as temperature or oxygen, thence, microencapsulation was supposed to protect these components [10,13].

The inhibition of the enzyme involved in the hydrolyzing carbohydrates such as *α*-amylase is important to approach for reducing hyperglycemia [14,15]. *Alpha*-amylase (E.C.3.2.1.1) is a potential protein tending to be a possibly applied inhibitor for anti-diabetic treatment. In human beings, *alpha*-amylase is a prominent enzyme which hydrolyses the *alpha* bonds of large, *alpha*-linked polysaccharides, such as starch and glycogen, yielding glucose and maltose [16,17,18]. The inhibition of *α*-amylase has minimized the absorption of glucose into the blood by delaying the digestion of carbohydrates. In recent years, several investigations have demonstrated the efficiency of *α*-amylase inhibitors in the treatment of diabetes mellitus such as Acarbose, Miglitol and Voglibose [17,18].

The spray-drying process has been one of the popular techniques for decades to encapsulate food ingredients because of its low operational cost and available equipment [19,20]. It has been used to produce powder from a liquid form by using carrier agent materials. Nowadays, maltodextrin is a popular carrier agent material to produce powder because it has many functionalities, such as wall materials, flavor carrier, bulking agents, reducing stickiness and improving product stability [21]. 

Therefore, this study was carried out to determine fragments containing bioactive compounds from various extracts (crude extract, petroleum ether extract, chloroform extracts and ethyl acetate extract) for spray-drying microencapsulation and evaluate the antidiabetic ability of spray-dried powder on in vitro α-amylase inhibitory activity test and the streptozotocin-induced diabetic mice model.

## 2. Materials and Methods

### 2.1. Chemicals and Reagents

The chemicals used including absolute ethanol, petroleum ether, chloroform, ethyl acetate, methanol, purchased from Chemsol Company, Vietnam. Aluminum chloride (AlCl_3_), and sodium carbonate (Na_2_CO_3_), Folin–Ciocalteu reagent, dimethyl sulfoxide, starch, and 5-dinitro salicylic acid (DNS), were purchased from Merck, Germany. Gallic acid, streptozotocin (STZ), nicotinamide (NAD), enzyme α-amylase, acarbose, p-nitrophenyl glucopyranoside (pNPG) were products of Sigma-Aldrich (St. Louis, MO, USA). The mouse was purchased from The Institute of Drug Quality Control–Ho Chi Minh City (IDQC–HCMC).

### 2.2. Plant Material

*E. hirta* was identified by the Institute of Tropical Biology under the Vietnam Academy of Science and Technology. The fresh *E. hirta* plant was harvested in Thu Duc District, Ho Chi Minh City, Vietnam, in October 2015. The whole plant without root was washed with tap water to remove all contamination and soaked in ethanol 70° to prevent the presence microorganisms. Then, the samples were dried in an oven at 60 °C for 8 h to remove the water content. After that, the sample was ground into powder and stored in a plastic bag for further uses.

### 2.3. Preparation of E. hirta Extracts

The plant powder (1.5 kg) was macerated with a methanol solvent by the ratio of 1:2 (*w/w*) in 2 weeks [22]. Then, the aqueous phase was filtered through Whatman No.1 filter paper, and the residue was added by fresh methanol. This process was repeated seven times until a clear colorless solution was obtained. Then, the extract solution was concentrated by a rotary evaporator at 60 °C to give 184.56 g of crude extract. Subsequently, the crude extract was mixed with distilled water by ratio 1:1 (*w/w*) and soaked with petroleum ether, chloroform, and ethyl acetate, respectively, to produce the fractional extract. The rotary evaporator was used to evaporate all the solvents to take the concentrated extracts. 

The percentage of the yield of the extract was calculated as
(1)$$\%\;\mathrm{Yield}\;\left(\%Y\right)=\frac{\mathrm{weight}\;\mathrm{of}\;\mathrm{dried}\;\mathrm{extract}\;\left(g\right)}{\mathrm{weight}\;\mathrm{of}\;\mathrm{sample}\;\left(g\right)}\times 100$$

### 2.4. Determination of Bioactive Compounds

#### 2.4.1. Determination of Total Phenolic Content (TPC)

The determination of the total phenolic content was based on the Folin–Ciocalteu assay method with some modification [23]. The reaction mixture consists of 0.2 mL of extract samples dissolving in methanol at 1 mg/mL. One milliliter of 10% Folin–Ciocalteu reagent was treated into the mixture and shaken well. After 5 min, 1.5 mL of 5% Na_2_CO_3_ solution was added into the mixture. Gallic acid was used as the standard solutions (20, 40, 40, 60, 80 and 100 μg/mL). The absorbance was measured at 750 nm by a spectrophotometer. The total phenol content was expressed as mg of gallic acid equivalent (GAE)/g of the extract [24].

#### 2.4.2. Determination of Total Flavonoid Content (TFC)

The total flavonoid content was determined by the spectrophotometric method [25]. The reaction mixture consists of 5 mL of extract samples dissolving in methanol at 0.4 mg/mL. Then, 5 mL of 2% AlCl_3_ solution was treated into the mixture. Rutin was used as a standard solution at different concentrations (20, 40, 40, 60, 80 and 100 μg/mL). The absorbance was measured at 415 nm by a spectrophotometer. The content of flavonoids in extracts was expressed in terms of rutin equivalent (mg of RU/g of extract) [24].

### 2.5. Spray-Drying Microencapsulation

Spray drying has been used to protect the ingredients that are sensitive to light, heat or oxygen. In this technique, a wall material protects the bioactive compounds [26,27]. The extracted sample was mixed with 20% maltodextrin at a ratio of 1:10, and the resulting mixtures were homogenized before spray dryer. The powder was obtained using LabPlant SD-06A spray dryer, serial number 485. The inlet temperatures were 180 °C, the outlet temperatures varied according to the inlet temperatures [28], the feed flow rate of the extracts was 10 rpm. Then, the *E. hirta* powder was used to determine the physical properties as the moisture content, color characteristic, buck density, solubility and hygroscopicity.

#### 2.5.1. Determination of Moisture Content

The moisture content was determined by moisture balance, type MOC-120H, No. D207302059. Triplicate samples of *E. hirta* powder (5 mg) were weighed and dried in the oven at 105 °C until its weight was constant [21].

#### 2.5.2. Color Characteristic

Color characteristic was measured by the Hue method. Distilled water-diluted triplicate samples of *E. hirta* powder to ratio 10:1 (*w/w*). The absorbance of this solution was measured at 510 nm and 610 nm and calculated the Hue:(2)$$\mathrm{Hue}=10\times \log\left(\frac{Abs_{510\mathrm{nm}}\;}{Abs_{610\mathrm{nm}}}\right)$$

The Hue Index usually ranges from 3 (a greenish-yellow or olive hue) to 7.5 (amber red-brown) for caramel colors.

#### 2.5.3. Solubility

The solubility of the samples was determined according to the reported procedures with some modification [29]. One gram of samples was added to 100 mL of distilled water, then the mixture was stirred at 600 rpm for 5 min. After that, 20 mL of the supernatant was transferred into petri-dishes and dried in an oven at 70 °C until the weight was constant. The solubility was calculated by weight difference and expressed in dry basis, considering the moisture content of each sample.

#### 2.5.4. Hygroscopicity

The hygroscopicity of the *E. hirta* powder was determined according to [28] with some modifications. The *E. hirta* powder was stored at room temperature in desiccators containing saturated sodium chloride solutions. The samples were weighed after one week, and the hygroscopicity was expressed in grams of the absorbed moisture per 100 g of dry solids.

### 2.6. In Vitro α-Amylase Inhibitory Activity

The α-amylase inhibitory activity was measured by the dinitrosalicylic acid method [16]. The 0.5 mL of the extract samples dissolving in dimethyl sulfoxide at different concentrations were pre-incubated with α-amylase 2 U/mL for 15 min. Then, 0.5 mL of the 1% *w/v* starch solution was added to the mixture, which was further incubated at 37 °C for 10 min. Then, the reaction was stopped by adding 1 mL DNS reagent and heated in a boiling water bath for 5 min. Acarbose was used as a positive control and the absorbance was measured at 540 nm. Percentage inhibition is calculated as
(3)$$\%\mathrm{Inhibition}=\frac{\left(Abs_{control}-Abs_{extract\;}\right)}{Abs_{control}}\times 100$$

The α-amylase inhibitory activity was expressed as the IC50 according to the percentage inhibition [30,31].

### 2.7. Acute Toxicity Testing

Spray powder safety evaluation was conducted by pyramiding single-dose (acute) toxicity testing [32]. A group of two mice (25–30 g) was fed with *E. hirta* powder with an increasing dosage on alternate days as 10, 30, 100, 300, 1000, 3000 and 5000 mg kg^−1^, with the dosing continuing until death. The toxicity test was found to be safe up to the dose 5000 mg/kg body weight; hence 1/10 of the dose was taken as an effective dose (500 mg/kg body weight).

### 2.8. Preliminary Anti-Hyperglycemic Test on STZ/NAD-Induced Mice

#### 2.8.1. Induction of STZ/NAD-Diabetes Mice

The model of the STZ/NAD-induced mice followed the guideline of Masiello et al. [33] with modification. Male mice weighing 22c25 received an intraperitoneally administered dose of 200 mg/kg bw of NAD, and after 15 min received another intraperitoneally administered dose of 100 mg/kg bw of STZ. After the intraperitoneal administration of STZ/NAD, the mice were observed for 30 min to examine for any abnormal signals. STZ/NAD-induced mice were fed with a daily special diet with fat milk and lard to make them obese. The blood glucose levels of mice were measured and recorded to check for the change in blood glucose. After 21 days, a model of diabetic mice was completed.

#### 2.8.2. Anti-Hyperglycemic Test

Diabetic mice (glucose level > 200 mg/dL) were divided into three groups of 6 mice each and orally administered as below:Group 1: negative control—orally administered with distilled water;Group 2: positive control—orally administered with acarbose (100 mg/kg bw);Group 3: sample group—orally administered with *E. hirta* powder (500 mg/kg bw).

The administration was continued for 15 days, once daily. Then, the blood glucose level was measured and recorded after 0, 1, 2 and 4 h on the first day and on days 1, 2, 6, 10 and 15 of administration.

### 2.9. Statistical Analysis

Statistical analysis was performed using SPSS 22.0. One-way variance analysis (one-way ANOVA) was applied to determine the significant difference between the samples at *p*-value < 0.05.

## 3. Results

### 3.1. *Euphorbia hirta* L. Extraction Yield

Nine and a half kilograms of fresh *E. hirta* produced around 1.5 kg of dried powder, 184.56 g of crude extract, 61.25 g of petroleum ether extract, 2.1 g of chloroform extract and 32.95 g of ethyl acetate extract. The results are shown in Figure 1. Different extracts were obtained from the *E. hirta* from the different solvents (petroleum ether, chloroform, and ethyl acetate) employed.

### 3.2. Phytochemical Analysis

According to the previous investigations, *Euphorbia hirta* L. contain a large amount of phenolic and flavonoid components [10]. In the present study, the total phenolic and flavonoid contents were shown in Table 1. The total phenolic content in the examined extracts was expressed in terms of gallic acid equivalent (mg of GAE/g extract) by the standard curve equation: y = 8.659x − 0.0153, R^2^ = 0.998. The value of the phenolic content was 194.55 ± 0.82, 51.85 ± 3.12, 81.56 ± 1.72 and 214.21 ± 2.53 mg of GAE/g extract for the crude extract, petroleum ether extract, chloroform extract and the ethyl acetate extract, respectively. The concentration of flavonoids was expressed as rutin (mg of RU/g extract) by the standard curve equation: y = 12.905x − 0.0515, R2 = 0.9888. The concentration of flavonoids in various plant extracts were 40.56 ± 7.27, 29.49 ± 1.66, 64.99 ± 2.60 and 91.69 ± 1.67 mg of RU/g extract. The highest phenolic and flavonoid concentrations were measured in ethyl acetate extract, and the lowest phenolic and flavonoid concentrations were measured in petroleum ether extract. The phenolic and flavonoid concentrations in the plant extracts depend on the solvent polarity which used in the extract preparation [24]. These data suggested that the total phenolic and flavonoid compounds were best extracted via ethyl acetate solvent from *E. hirta*. Therefore, the ethyl acetate extract was suggested to further investigate and determine the bioactive compounds and identify their antidiabetic activity. 

### 3.3. Spray-Drying Microencapsulation

The physical properties of spray powder were shown to clarify the quality of the *E. hirta* powder (Table 2). The moisture content effects on the shelf-life of dried materials, and the shelf-life of powder can extend in moisture under than 10% [34]. The moisture content in the *E. hirta* powder is around 4.9567 ± 0.00577%, which is less than 5% so that it is suitable for storage for a long time. Bulk density is a property of food powder, the bulk density of *E. hirta* powder ranges from 0.36 to 0.48 g/mL, and the high bulk density increases its sticky or less free-flowing nature. In this research, the *E. hirta* powder had a bulk density of 0.3715 ± 0.01286 g/mL, as it adapted the requirement. Solubility shows the ability of the powder to be dissolved in water, and the solubility of the *E. hirta* powder is around 95.83 ± 1.44%. Color is a major quality parameter in a dried food product; color was presented in the Hue Index value. Table 2 shows that the Hue index value around 3.7367 ± 0.1424 means that the color of spray sample is yellow. Hygroscopicity is the ability of food powder to absorb moisture from a high relative humidity environment. The result from Table 2 showed that *E. hirta* powder could absorb 9.9890 ± 1.4538 g H_2_O in a 100 g sample.

### 3.4. In Vitro α-Amylase Inhibitory Activity Essay

The inhibition of α-amylase minimized the absorption of glucose into blood by the delay of the digestion of carbohydrates. The in vitro α-amylase inhibitory activities of the *E. hirta* powder were assayed. The result of this study showed that spray powder has a good α-amylase inhibitory activity (Figure 2) with the IC50 value as evidence (Table 3). The *E. hirta* powder and acarbose—a standard drug in diabetes management (at concentration 10 mg/mL) showed a 51.19% and 94.69% inhibitory effect on α-amylase, respectively. The IC50 value of the *E. hirta* powder was 5.725 mg/L, meanwhile for acarbose it was 2.511 mg/mL. Although the α-amylase inhibitory activity of *E. hirta* powder is not strong enough as that of acarbose is, it also proved that *E. hirta* can inhibit the carbohydrate-hydrolyzing enzyme and has potential antidiabetic activity.

### 3.5. Acute Toxicity

The mice orally administered 10, 30, 100, 300, 1000, 3000, 5000 mg kg^−1^ doses of *E. hirta* powder were kept under observation for two weeks. After two weeks, all the mice were alive and did not show any toxic symptoms such as body weight loss. Therefore, it was found that 5000 mg kg^−1^ dose of the powder showed a confidence dose and was considered as safe. Therefore, it was concluded that the median lethal dose (LD50) was more than 5000 mg kg^−1^ when administered orally (Table 4). Therefore, the extracts used in this study are safe for long administration.

### 3.6. Anti-Hyperglycemic Test

This experiment was undertaken to evaluate the hypoglycemic activity of the spray powder containing *Euphorbia hirta* L. in the STZ-induced diabetic mice. Diabetic mice usually have fast blood glucose levels above 200 mg/dL. Acarbose is commonly used as a standard antidiabetic drug in STZ-induced diabetes to compare the anti-hyperglycemic extent of bioactive compounds. In normoglycemic rats, the test showed a significant reduction of blood glucose level till the end of 4 h.

The fast blood glucose levels in 4 h showed that after 4 h of treatment with the *E. hirta* powder (Figure 3), there was a 51% reduction in the fast blood glucose level, whereas the treatment with acarbose and distilled water at the same time produced a 44% and 35% reduction. In a short time, the *E. hirta* powder reduced fast blood glucose levels better than distilled water and acarbose.

A significant reduction of 23.32% in fast blood glucose was observed after 15 days of treatment with the *E. hirta* powder, compared to the 30.87% and 16.89% reduction when treated with acarbose and distilled water, respectively (Figure 4). In the longer time treatment, acarbose was the competitive inhibitor in the reduction of fast blood glucose levels, however, the reduction of the fast blood glucose levels of the *E. hirta* powder was close to acarbose and better than distilled water. Therefore, the result of this test showed that the *E. hirta* powder exhibited a significant reduction in the fast blood glucose levels.

## 4. Discussion

Plants are a rich resource of natural bioactive compounds, such as phenolics, flavonoids and their derivatives. These compounds have received attention because of their bioactivities and physiological functions, including antioxidant, anti-allergic, anti-inflammatory, antimicrobial, and antidiabetic activities [35]. From the crude extract of *Euphorbia hirta* L. containing multiple different groups of bioactive compounds, liquid–liquid extractions can be employed to separate many different fractions. During this procedure, the chemical constituents of the extract are separated based on their polarity by specific solvents including petroleum ether, chloroform, and ethyl acetate. This procedure typically results in a total of four extracts. The qualitative phytochemical analysis in *E. hirta* was obtained from these extracts. In relation to the solvent used, high concentrations of the total phenolic content and total flavonoid content were found in the chloroform and ethyl acetate extracts. The value of the phenolic contents was 81.56 ± 1.72 and 214.21 ± 2.53 mg of GAE/g extract for the chloroform extract and the ethyl acetate extract, respectively. The concentration of flavonoids was 64.99 ± 2.60 and 91.69 ± 1.67 mg of RU/g extract for the chloroform extract and the ethyl acetate extract, respectively.

In the previous studies, the point of focus has been on elucidating the mechanism of action and the phytochemicals of plant extracts for traditional diabetes treatment. The study of the inhibitory activity of phytochemicals against α-amylase and α-glucosidase has been widely popular. Several phytochemicals including flavonoids and phenolics have been detected. The presence of these compounds, especially in chloroform and ethyl acetate extracts, demonstrated ability in the treatment of diabetes. The inhibition of the enzymes involved in the metabolism of saccharides such as α-amylase is an important therapeutic strategy for reducing hyperglycemia [14]. Polyphenolic compounds have popularly reported the inhibitory activity against α-amylase in both in vitro and in vivo experiments [11,12,36]. Phenolics and flavonoids found in *E. hirta* such as quercetin, quercitrin, and rutin were proved to be effective inhibitors of mammalian α-amylase. The *E. hirta* powder and acarbose (at concentration 10 mg/mL) showed 51.19% and 94.69% inhibitory effects on α-amylase, respectively. The IC50 value of the *E. hirta* powder was 5.725 mg/L, meanwhile acarbose was 2.511 mg/mL. Although the α-amylase inhibitory activity of *E. hirta* powder was not strong enough as acarbose, it also proved that *E. hirta* had a mild α-amylase inhibition and has potential antidiabetic activity. This result is consistent with the reported investigation [37,38].

The control of postprandial plasma glucose concentrations is the key to the the treatment of diabetes mellitus and its related complications. The results of this study showed that *Euphorbia hirta* L. powder from the highest number of bioactive compounds possessed antidiabetic activities. Phenolics and flavonoids are commonly distributed in plants and have been demonstrated to possess potential effects in the treatment of diabetes disease. The possible mechanism of antidiabetic action of *E. hirta* is the inhibition of α-amylase and the reduction of the fast blood glucose level on diabetic mice [39]. Therefore, this study supports the drug/functional food development from the *E. hirta* extract in the management of diabetes.

## 5. Conclusions

This study provided significant evidence for the antidiabetic activity of Euphorbia hirta L. One of the possible mechanisms of antidiabetic activity of this plant is related to the inhibitory action of the key enzymes involving the hydrolysis of carbohydrates, to control the blood glucose concentration in the body. Recent investigations have proved that phenolics and flavonoids from natural resources containing bioactive components have hypoglycemia in action. Therefore, in further studies, we will continue to investigate the stabilities in functional food application. Besides, the fragments of *E. hirta* extracts having antidiabetic activities will be carried out to isolate and identify the bioactive compounds for the in vitro and in vivo studies of their antidiabetic activity. Furthermore, it is necessary to elucidate the mechanisms of action of the extracts and phytochemicals of this plant at the cellular and molecular levels.

## Figures and Tables

**Figure 1 foods-09-00881-f001:**
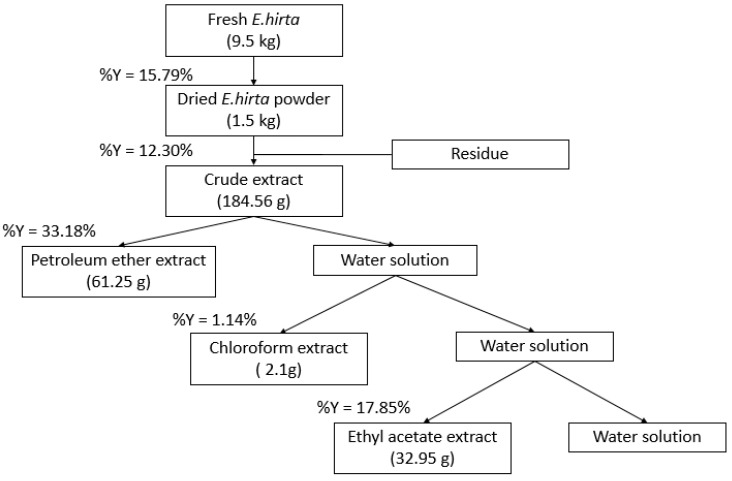
Percentage of extracts from *Euphorbia hirta* L.

**Figure 2 foods-09-00881-f002:**
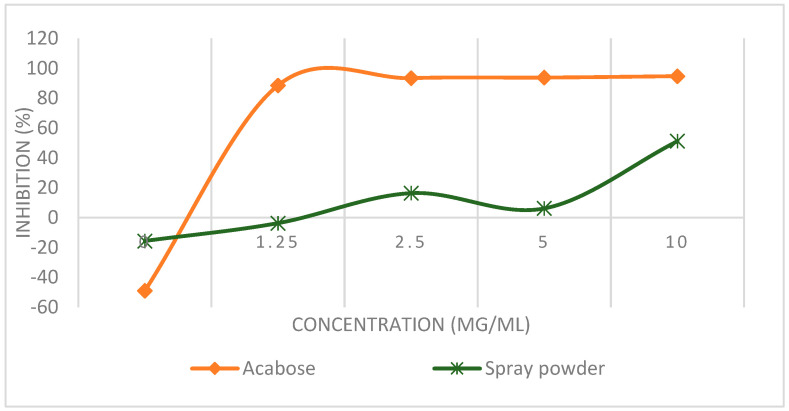
The inhibitory potency of the *E. hirta* powder against α-amylase activity. Each value is the average of three analyses ± standard deviation (*n* = 3).

**Figure 3 foods-09-00881-f003:**
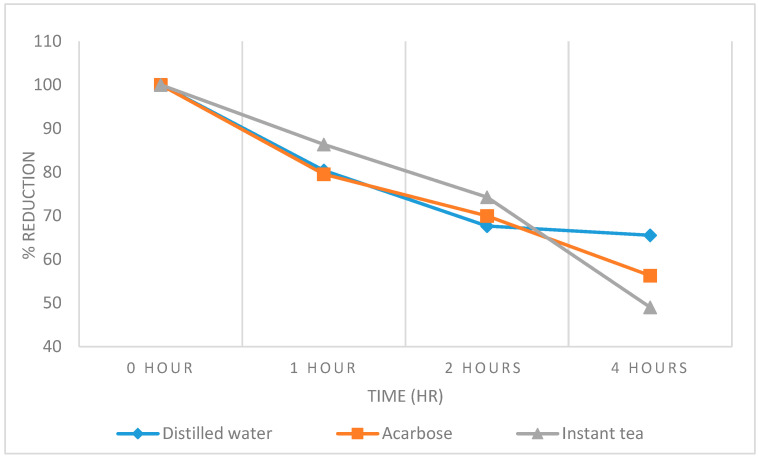
Effect of the *E. hirta* powder on the streptozotocin-induced diabetic mice in 4 h. Values are presented as the means ± standard deviation (*n* = 6).

**Figure 4 foods-09-00881-f004:**
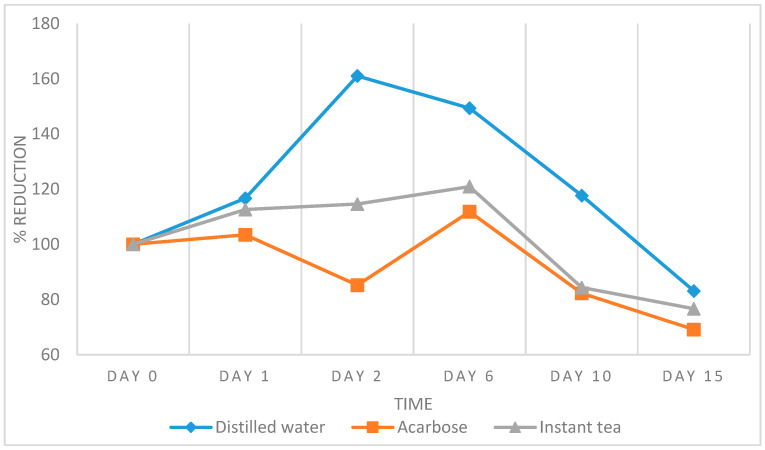
Effect of the *E. hirta* powder on the streptozotocin (STZ)-induced diabetic mice in 15 days. Values are presented as the means ± standard deviation (*n* = 6).

**Table 1 foods-09-00881-t001:** Total phenolic and flavonoid contents in the *E. hirta* extracts.

Extract	Total Phenolic Content (mg of GAE/g Extract)	Total Flavonoid Content (mg of RU/g Extract)
Crude extract	194.55 ± 0.82 ^1^	40.56 ± 7.27
Petroleum ether extract	51.85 ± 3.12	29.49 ± 1.66
Chloroform extract	81.56 ± 1.72	64.99 ± 2.60
Ethyl acetate extract	214.21 ± 2.53	91.69 ± 1.67

^1^ Each value is the average of three analyses ± standard deviation (*n* = 3).

**Table 2 foods-09-00881-t002:** Physical properties of the spray power.

Yield (%)	Moisture (%)	Bulk Density (g/mL)	Solubility (%)	Hygroscopicity (g H_2_O/100 g)	Hue Index
21.034	4.9567 ± 0.00577 ^1^	0.3715 ± 0.01286	95.83 ± 1.44	9.9890 ± 1.4538	3.7367 ± 0.1424

^1^ Each value is the average of three analyses ± standard deviation (*n* = 3).

**Table 3 foods-09-00881-t003:** The IC50 of the *E. hirta* powder on α-amylase inhibition.

Sample	α-Amylase Equation (y = ax + b)	α-Amylase IC50 (mg/mL)
Acarbose	y = 29.246x − 23.446	2.511
Spray powder	y = 14.355x − 32.175	5.725

**Table 4 foods-09-00881-t004:** Pyramiding dose.

Dosage	Mortality
10 mg kg^−1^	0/2
30 mg kg^−1^	0/2
100 mg kg^−1^	0/2
300 mg kg^−1^	0/2
1000 mg kg^−1^	0/2
3000 mg kg^−1^	0/2
5000 mg kg^−1^	0/2
Both minimal lethal dosage and LD50 > 5000 mg kg^−1^	

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
