# Peer review of "Spray-Drying Microencapsulation of High Concentration of Bioactive Compounds Fragments from Euphorbia hirta L. Extract and Their Effect on Diabetes Mellitus"

_foods, 2020, doi:10.3390/foods9070881_

Round 1

Reviewer 1 Report

Comments

The manuscript “Spray-drying microencapsulation of high concentration of bioactive compounds fragments from Euphorbia hirta Linn. extract and their effect on diabetes mellitus.” describes the microencapsulation (spray-drying) of bioactive compounds from Euphorbia hirta Linn. to obtain powder which has antidiabetes potential.

Remarks

1- The manuscript has many of grammatical errors.

2- Check the name of the taxonomic authority: Euphorbia hirta L. is the accepted name (The plant liste). The authors have to change this in all the manuscript.

3- Introduction should be enriched with recent references (2017-2020).

4-In the discussion section, the authors gave a short discussion. Moreover, the authors must give and discuss the interesting values of their findings and compare them with the previous studies. This section should be rewritten.

5-The references must be revised according to the style of the journal.

Some suggested corrections

Abstract

Page 1, Lines 15 and 16: …… content (TPC) and total flavonoid content (TFC) of three different extracts (petroleum ether extract, chloroform extract and ethyl acetate extract) from a crude extract of Euphorbia hirta  ………[There is a confusion because the authors performed the TPC and TFC on four extracts].

Page 1, line 23:…. Ethyl acetate extract fraction was mixed….[Please delete fraction]

Page 1, line 29:….  glucose level in STZ diabetic mice…..[Give the full name of STZ].

Introduction

Page 2, line 55: …. to Euphorbia genus of Euphorbiaceae family…..change to ….. to Euphorbia genus of Euphorbiaceae family. ….[family not in Italic]

Plant Material

Page 2, line 87: …The fresh E.hirta plant ….shoild be …. The fresh E. hirta plant ……

Page 3, line 92: ……. 2.3. Preparation of E.hirta extracts ….should be …. 2.3. Preparation of E. hirta extracts ……….

Page 3, line 93 : …. The plant powder [give the amount] was macerated with methanol solvent by the ratio of 1:2 [precise the ratio] in 2 weeks.

Page 3, lines 95-97: …. Then, the extract solution was concentrated by a rotary evaporator at 600C, the concentrated extract called crude extract. …..change to …. Then, the extract solution was concentrated by a rotary evaporator at 600C to give ….ml of crude extract. …..

Page 3, line 109:…. as mg of GAE/gm of ….should be …. as mg of GAE/g of ….

Page 4, line138: …. 1 gram of samples was ….change to …. 1 g of samples was …..

Page 5, line 178:…. Orally administered with E. hirta powder ….modify to …. Orally administered with E. hirta powder …………..

Page 6, lines 209 and 217: ….table 1 …..should be ….table 2

Page 7, lines 235 and 238:….Acabose ….should be ….Acarbose …..

Page 8, lines 261 and 262: … that E. hirta powder a significant reduction …..change to ….. that E. hirta powder exhibits a significant reduction ……… 

Conclusion

Page 9, line 276: … of Euphorbia Hirta ….should be ….. of Euphorbia hirta …………..  

Author Response

Remarks

  • The manuscript has many of grammatical errors. à The English had been revised. We try to fix errors as much as possible.
  • Check the name of the taxonomic authority: Euphorbia hirta L. is the accepted name (The plant list). The authors have to change this in all the manuscript. à The name of this plant had been revised.
  • Introduction should be enriched with recent references (2017-2020). à The references had been revised.
  • In the discussion section, the authors gave a short discussion. Moreover, the authors must give and discuss the interesting values of their findings and compare them with the previous studies. This section should be rewritten. à The discussion section had been revised.
  • The references must be revised according to the style of the journal. à The references had been revised.

Some suggested corrections

Abstract

Page 1, Lines 15 and 16: …… content (TPC) and total flavonoid content (TFC) of three different extracts (petroleum ether extract, chloroform extract and ethyl acetate extract) from a crude extract of Euphorbia hirta ………[There is a confusion because the authors performed the TPC and TFC on four extracts]. à The confusion had been revised.

Page 1, line 23:…. Ethyl acetate extract fraction was mixed….[Please delete fraction] à The mistake had been revised.

Page 1, line 29:…. glucose level in STZ diabetic mice…..[Give the full name of STZ]. à The full name had been given.

Introduction

Page 2, line 55: …. to Euphorbia genus of Euphorbiaceae family…..change to ….. to Euphorbia genus of Euphorbiaceae family. ….[family not in Italic] à The mistake had been revised.

Plant Material

Page 2, line 87: …The fresh E.hirta plant ….should be …. The fresh E. hirta plant …… à The mistake had been revised.

Page 3, line 92: ……. 2.3. Preparation of E.hirta extracts ….should be …. 2.3. Preparation of E. hirta extracts ………. à The mistake had been revised.

Page 3, line 93 : …. The plant powder [give the amount] was macerated with methanol solvent by the ratio of 1:2 [precise the ratio] in 2 weeks. à The amount of powder and precise ratio had been added.

Page 3, lines 95-97: …. Then, the extract solution was concentrated by a rotary evaporator at 600C, the concentrated extract called crude extract. …..change to …. Then, the extract solution was concentrated by a rotary evaporator at 600C to give ….ml of crude extract. …à Revised

Page 3, line 109:…. as mg of GAE/gm of ….should be …. as mg of GAE/g of …à Revised

Page 4, line138: …. 1 gram of samples was …change to …. 1 g of samples was …. à Revised

Page 5, line 178:…. Orally administered with E. hirta powder ….modify to …. Orally administered with E. hirta powder ………….. à The mistake had been revised.

Page 6, lines 209 and 217: ….table 1 …..should be ….table 2 à The mistake had been revised.

Page 7, lines 235 and 238:….Acabose ….should be ….Acarbose ….. à The mistake had been revised.

Page 8, lines 261 and 262: … that E. hirta powder a significant reduction …..change to ….. that E. hirta powder exhibits a significant reduction ……… à The mistake had been revised.

Conclusion

Page 9, line 276: … of Euphorbia Hirta ….should be ….. of Euphorbia hirta … à The mistake had been revised.

Reviewer 2 Report

Dear Authors,

This work is very interesting, but I have some comments. Namely, line 186: I suggest to change the subtitle of section 3.1 to: "E. hirta extraction yield"
In my opinion, the extracts should be more thoroughly phytochemically tested qualitatively and quantitatively. This can be done quickly by the UHPLC-MS-qTOF method. The discussion should include a comparison of the results obtained with literature data, if any, especially in respect to in vitro α‑amylase inhibitory activity essay, acute toxicity, anti-hyperglycemic test. These activities can be compared to activities in other species of the genus Euphorbia, if studied. You must compare the results obtained with the reference values ​​in the literature to correctly evaluate the activity of the material you are tested.

Author Response

In my opinion, The extracts should be more thoroughly phytochemically tested qualitatively and quantitatively. This can be done quickly by the UHPLC-MS-qTOF method. The discussion should include a comparison of the results obtained with literature data, if any, especially in respect to in vitro α‑amylase inhibitory activity essay, acute toxicity, anti-hyperglycemic test. These activities can be compared to activities in other species of the genus Euphorbia, if studied. You must compare the results obtained with the reference values ​​in the literature to correctly evaluate the activity of the material you are tested.

Line 186: change the subtitle of section 3.1 to: "E. hirta extraction yield" à The subtitle had been revised.

The extracts should be more thoroughly phytochemically tested qualitatively and quantitatively. à Thank you for your suggestion. We do plan for this test in further study.

The discussion had been revised by comparing the result of our study with literature data.